# Efficacy of Chitosan Nanoparticle Loaded-Salicylic Acid and -Silver on Management of Cassava Leaf Spot Disease

**DOI:** 10.3390/polym14040660

**Published:** 2022-02-09

**Authors:** Nguyen Huy Hoang, Toan Le Thanh, Wannaporn Thepbandit, Jongjit Treekoon, Chanon Saengchan, Rungthip Sangpueak, Narendra Kumar Papathoti, Anyanee Kamkaew, Natthiya Buensanteai

**Affiliations:** 1School of Crop Production Technology, Institute of Agricultural Technology, Suranaree University of Technology, Nakhon Ratchasima 30000, Thailand; huyhoangqct@gmail.com (N.H.H.); w.thepbandit@gmail.com (W.T.); c.saengchan5310@gmail.com (C.S.); fongfangfang_m5430222@hotmail.com (R.S.); narendrakumar.papathoti@gmail.com (N.K.P.); 2Department of Plant Protection, College of Agriculture, Can Tho University, Can Tho 900000, Vietnam; lttoan@ctu.edu.vn; 3School of Chemistry, Institute of Science, Suranaree University of Technology, Nakhon Ratchasima 30000, Thailand; yuiyongyuy@gmail.com (J.T.); anyanee@g.sut.ac.th (A.K.)

**Keywords:** cassava leaf spot, chitosan, ionic gelation method, nanoparticle, salicylic acid, silver

## Abstract

Leaf spot is one of the most important cassava diseases. Nanotechnology can be applied to control diseases and improve plant growth. This study was performed to prepare chitosan (CS) nanoparticle (NP)-loaded salicylic acid (SA) or silver (Ag) by the ionic gelation method, and to evaluate their effectiveness on reducing leaf spot disease and enhancing the growth of cassava plants. The CS (0.4 or 0.5%) and Pentasodium triphosphate (0.2 or 0.5%) were mixed with SA varying at 0.05, 0.1, or 0.2% or silver nitrate varying at 1, 2, or 3 mM to prepare three formulations of CS-NP-loaded SA named N1, N2, and N3 or CS-NP-loaded Ag named N4, N5, and N6. The results showed that the six formulations were not toxic to cassava leaves up to 800 ppm. The CS-NP-loaded SA (N3) and CS-NP-loaded Ag (N6) were more effective than the remaining formulations in reducing the disease severity and the disease index of leaf spot. Furthermore, N3 at 400 ppm and N6 at 200, 400, and 800 ppm could reduce disease severity (68.9–73.6% or 37.0–37.7%, depending on the time of treatment and the pathogen density) and enhance plant growth more than or equal to commercial fungicide or nano-fungicide products under net-house conditions. The study indicates the potential to use CS-NP-loaded SA or Ag as elicitors to manage cassava leaf spot disease.

## 1. Introduction

The cassava (*Manihot esculenta* Crantz) plant and its tapioca are an important food source for world food security, especially in developing countries. Cassava is used in the production of food, industry, and animal feed [1]. Thailand’s cassava acreage and production reached 1.34 million hectares and 30.84 million tons in 2017, which has increased 1.15-fold over the previous ten years. Although Thailand’s acreage and production account for only 5.45% and 11.04% of the world, respectively, the quantity and value of Thailand’s cassava crop are between 58.5–81.2% and 44.4–56.7% in the world export market, respectively. Furthermore, cassava is Thailand’s key export crop [2]. Cassava can tolerate drought or nutrient-poor soil, so it also has a role in water-deficient farming areas. However, the biotic stress on a living organism including plant diseases, insects, and weeds, or abiotic stress, such as adverse environmental factors, could affect the growth and development of cassava, resulting in loss of yield. Currently, twenty-eight types of cassava diseases caused by fungi, viruses, or bacteria have been recorded [3,4]. Leaf spot disease is one of the most important cassava diseases. The disease causes a loss of up to 30% of cassava yield. However, the serious problem of cassava leaf spot disease has often been neglected, until a recent outbreak in Brazil [5]. In 2019, *Alternaria* sp. was reported as a pathogen causing leaf spot disease on cassava [6]. It is now possible to implement cultural methods, chemical methods, and host resistance to achieve effective control over or manage diseases. Fungicides containing copper, benomyl, thiophanate, carbendazim, flutriafol, cyproconazole, pyraclostrobin, thiophanate-methyl, tebuconazole, and azoxystrobin can control pathogens with varying degrees of effectiveness. Hence, fungicides are a popular method to reduce the damage of cassava leaf spot. Also, the cassava cultivars Sri Prakash and Sri Visakam are recommended for their resistance to cassava brown leaf spot disease in India [5,7,8]. However, this leaf spot disease-resistant cultivar is not present in Thailand. Therefore, it is necessary to look for a new method to increase the resistance of cassava. Elicitation by stimulating the secondary metabolites is one of the most effective tools to enhance plant immunity. In general, the plant has an innate immune system against adverse environmental factors, including abiotic and biotic stresses, which differs depending on the cultivar and the adverse factors. The plant’s immune system can be artificially induced by an elicitor. Elicitors are biotic or abiotic compounds that activate defense mechanisms and innate immunity in plants against pathogens and stress conditions. The elicitors may be chemical, microbial, chitosan (CS), plant extracts, algal extracts, composts, or biochar. As such, applying appropriate elicitors could aid plants against pathogens as well as cassava leaf spot disease [9,10].

In recent years, nanotechnology has been applied to many fields in agriculture, including nanofertilizers, nanobiotechnology, nanomaterials, nanosensors, nanopesticides, nanoelicitors, and nanoherbicides, to enhance plants’ tolerance to biotic and abiotic stress, improve crop yield and quality, and especially to build sustainable agriculture [11,12,13,14]. Nanoparticles (NPs) are used as protectants (silver, gold, copper, titanium dioxide, and CS) or carriers (CS, silica, solid lipid, and layered double hydroxide) of active compounds (insecticides, fungicides, herbicides, and RNA-interference) to protect plants against bacteria, fungi, viruses, and insects. The advantages of applying pesticide-based nanoparticles in agriculture include improved shelf-life, target site-specific uptake, increased solubility, and reduced soil leaching and toxicity [15]. NPs in the form of nutrients and non-nutrients are provided to plants through leaves or roots to improve plant health as well as control plant diseases. In many recent reviews, NPs are considered a biosafe technique. On the indirect side, the amount of chemical pesticides or fertilizers used for crop production is reduced because it is replaced by NPs (nano fertilizers, nano pesticides, and nano elicitors). From a direct perspective, applying NPs to soil may have a negative impact on microbial communities, but to a low degree when compared to chemical application. Although the risk is low, the potential toxicity and hazardous effects also deserve attention. Usually, the safety-by-design principle is applied to screen the potential risks of materials and methods of synthesis to NP formulation [16,17,18,19]. CS is a natural polysaccharide with superior characteristics, including low toxicity, affordability, biodegradability, biocompatibility, environmental non-toxicity, and absorption abilities that have been applied in crop production for plant disease management and enhancing crop yields [20,21]. The biogenic Ag-NPs are environmentally safer, with more interest as high-potential antifungal and antibacterial agents [22]. Foliar spray of Ag-NPs at 50–70 ppm on tomato plants reduced the severity of diseases caused by *Tomato mosaic virus* and *Potato virus Y* 3.9–4.8- and 2.2–4.5-fold, respectively. It also increased the chlorophyll content, total soluble protein, activities of peroxidase (POD), and polyphenol oxidase (PPO) [23]. Also, CS-NP-loaded salicylic acid (SA) at 0.01–0.16% can inhibit *Fusarium verticillioides* mycelium growth by 62.2–100% and spore germination by 48.3–60.5% in in vitro conditions. In addition, it acted as an elicitor and was able to activate the defense system of the maize plants to reduce post-flowering stalk rot disease by 40.5 to 59.47% and increase yields 1.3–1.5-fold when compared with the control in field conditions [24]. CS-NP-loaded Cu at 0.1% has been able to inhibit mycelium growth of *Alternaria alternata*, *Macrophomina phaseolina*, and *Rhizoctonia solani* by 89.5, 63.0, and 60.1%, which was higher than CS-NP-loaded saponin and CS-NP was 80.9, 66.2, 27.7% and 82.2, 87.6, 34.4%, respectively. In addition, both CS-NP, CS-NP-loaded Cu, and CS-NP-loaded saponin at 0.06% inhibited the germination of *A. alternata* by 84.4, 83.3, and 78.3%, respectively [25]. There are two approaches to synthesizing NPs, including top-down and bottom-up methods, which result in NPs with different sizes, shapes, and functions. Some synthetic nanoparticles used as a pesticide are not too difficult to implement, such as sol-gel processes, green synthesis by microorganisms, or plant extracts [26]. The ionic gelation technique for the production of micro-particles or NPs is based on the electrostatic interaction between ions with different charges; it was discovered by Calvo et al. in 1997 [27,28]. This system can load additional macromolecules or drugs as a delivery system to improve biological activity or efficiency. The method is simple, fast, economical, easy to implement, and does not use organic solvents, but it is important to select materials and optimize the process to produce suitably effective NPs [29,30,31,32,33,34].

Control studies of treating cassava brown leaf spot with synthetic pesticides were performed. However, the application of NPs on cassava plants to control or manage cassava diseases in general, and cassava leaf spot in particular, is not yet available. Moreover, ionic gelation is an easy and environmentally friendly method of NP production if the materials are properly selected and the process is optimized. In this study, CS-NP-loaded SA or Ag was prepared by ionic gelation method. Then, their effectiveness as nanoelicitors to reduce leaf spot disease and enhance the growth of cassava plants was evaluated in net-house conditions. More specifically, this study is approached with a focus on the reverse research model of preparing elicitors, toxicity tests, screening formulation effectiveness to concentration effectiveness, and characterizing effective elicitors.

## 2. Materials and Methods

### 2.1. Materials

Chitosan low molecular weight (Sigma-Aldrich, Dublin, Iceland), Penta-Sodium triphosphate (Merck, Darmstadt, Germany), SA (HiMedia, Mumbai, India), and silver nitrate (Sigma-Aldrich, Dublin, Iceland) were provided by the School of Chemistry, Institute of Science, Suranaree University of Technology, Thailand. The pathogen *A. alternata* strain H-Vi 7 was provided by the Plant Pathology & Biopesticide Laboratory, Suranaree University of Technology, Thailand. The fungi were cultured from Eppendorf stock on potato dextrose agar (PDA) medium (200 g potato extract; 20 g dextrose, 18 g agar, 1 L distilled water) at 27 ± 2 °C for 2 days. Then, the mycelium was transferred to a new PDA and incubated until the mycelium grew to the edge of the Petri plate [35,36]. Then, the surface colony was streaked by a sterile needle to enhance conidia production. A total of 5 mL of sterilized distilled water was added to each Petri plate to harvest conidia. The mixture was filtered through fabric to remove mycelium. The concentration of the suspension was determined using a hemocytometer and adjusted to 1 × 10^4^ or 1 × 10^5^ conidia mL^−1^ by adding sterilized distilled water [35].

### 2.2. Synthesis of Chitosan Nanoparticles Loaded-Salicylic Acid and -Silver

CS-NP-loaded SA was synthesized according to the description of [24] with minor modifications. In brief, CS (0.4% *w*/*v*) was dissolved in 1% acetic acid in 500 mL of distilled water by stirring at 300 rpm overnight, which was followed by filtering through Whatman paper 1 with a particle retention of 11 µm (90 mm of diameter). TPP (0.2% *w*/*v*) was prepared in 500 mL of distilled water. In addition, SA was prepared at various concentrations including 0.05%, 0.1%, and 0.2% (*w*/*v*) in 500 mL of distilled water. While the CS solution was stirred at 400 rpm, TPP and SA were added to each by a syringe. The mixed system was maintained at 600 rpm for 8 h. An equal volume ratio between CS, TPP, and each SA concentration including 0.05%, 0.1%, and 0.2% was combined to formulate 3 types of CS-NP-loaded SA, which were coded as N1, N2, and N3, respectively.

The CS-NP-loaded Ag was synthesized with a CS:TPP mass ratio of 1:1 based on the description of [37] and [38], with modification. In brief, 500 mL of CS (0.5% *w*/*v*) and TPP (0.5% *w*/*v*) were prepared in the same way as CS-NP-loaded SA. A total of 500 mL of silver nitrate 1, 2, and 3 mM was prepared in distilled water. The three types of CS-NP-loaded Ag with different concentrations of silver nitrate, including 1, 2, and 3 mM, were synthesized similarly to the CS-NP-loaded SA and were coded as N4, N5, N6, respectively. Each formation of NP was harvested by centrifuging the mixture at 9500× *g* rpm at 4 °C for 15 min. The pellets were collected and freeze-dried, then stored at 4 °C until used.

### 2.3. Characterization of Elicitor- NPs

The particle size, zeta potential, and PDI (weight average molecular weight per number average molecular weight) of N3 and N6 were measured by Zetasizer Nano ZS (Malvern Instruments Ltd., Worcestershire, UK) at Suranaree University of Technology, Thailand. Similarly, the morphology and size and interaction groups of N3 and N6 were detected by a Field Emission Scanning Electron Microscope (FESEM, Carl Zeiss AURIGA^®^ CrossBeam^®^ Workstation, Carl Zeiss Microscopy GmbH, Jena, Germany) and FTIR (Bruker Tensor 27 FT-IR spectrophotometer, Bruker Optics Ltd., Ettlingen, Germany), respectively. In FTIR analysis, the NPs N3 and N6 and freeze-dried or bulk CS, were finely ground with KBr with the ratio 1:99. Then, the KBr pellet was inserted into the IR sample holder. The spectra were collected with a transmission mode in the range of 400 and 4000 cm^−1^ wavelengths. The peaks were collected by scanning 32 times and analysis by OPUS 7.5 (Bruker Optics Ltd., Ettlingen, Germany) [39].

### 2.4. Phytotoxicity Test

A phytotoxicity test was conducted to assess the toxic potential of NPs on cassava using the leaf disk assay method according to the descriptions of [40,41]. The disks of mature cassava leaf blades were prepared using a cork borer with a diameter of 8 mm. Formulated NPs, including N1, N2, N3, N4, N5, and N6, were prepared with distilled water into a series of solutions at 25, 50, 100, 200, 400, and 800 ppm. A disk was immersed in 1 mL of NP solution. Fungicides including Headline^®^ (Pyraclostrobin), JOINT^®^ (Flutriafol), and ZONO-S1^®^ (Zinc oxide NP) at 10 mL, 30 mL, and 20 mL per 20 L (recommended dose) were used as positive controls that were coded as Pyr, Flu, ZON, respectively. Distilled water was used as a negative control. SA, CS, and SN at 100 ppm were also used as a control group (Table 1). The symptoms of the leaf disks were observed visually after they were washed with distilled water at 24 h after incubation. The assessment scale consists of 4 levels: 0, non-effect; 1, an area of necrotic spots <50%; 2, an area of necrotic spots 50–70%; 3, an area of necrotic spots 70–90%; and 4, an area of necrotic spots >90%.

### 2.5. Screening Elicitor Formulations for Inducing Resistance against Cassava Leaf Spot Disease and Growth in Cassava Plants under Net-House Conditions

The experiment was carried out in a completely randomized design (CRD) with four replications. The stalks of cassava variety Pirun 2 were soaked for 5 min with NPs solutions at 25, 50, 100, 200, 400, and 800 ppm, water (negative control), fungicides (positive control), and CS, SA, and silver nitrate at 100 ppm (control group), as described in Table 1, before planting in a pot containing sandy soil, which was kept in a net-house. At 28 and 42 days after plating (DAP), the cassava plants were treated with elicitors by foliar spray, 5 mL per plant. At 44 DAP, the cassava plants were inoculated by spraying the suspension of *A. alternata* strain H-Vi 7 at 1 × 10^4^ conidia mL^−^^1^. They were covered with plastic and sprayed with water to create a relative humidity (>80%) condition [24,35,42,43]. The plants were kept in a net-house to monitor disease symptoms. The disease score was assessed at 12 days after inoculation based on the diagrammatic scale (0 to 8) following the description of [44]. Disease severity (DS) was calculated according to the following formula (1):(1)$$\mathrm{DS}\;(\%)=\frac{\mathrm{Sum}\;\mathrm{of}\;\mathrm{all}\;\mathrm{numerical}\;\mathrm{scoring}\;}{\mathrm{The}\;\mathrm{number}\;\mathrm{of}\;\mathrm{leaves}\;\times \;\mathrm{Maximum}\;\mathrm{score}}\;\times \;100$$

Disease index (DI) was calculated according to the following formula (2):(2)$$\mathrm{DI}\;(\%)=\frac{\mathrm{The}\;\mathrm{number}\;\mathrm{of}\;\mathrm{leaves}\;\mathrm{appeared}\;\mathrm{symptom}\;}{\mathrm{The}\;\mathrm{total}\;\mathrm{leaves}}\;\times \;100\;$$

The DS and DI were analyzed statistically according to two factors by SPSS software version 20, including the 6 NPs formulated and the 6 concentrations to select the most effective CS-NP-loaded SA and CS-NP-loaded Ag. Then, the DS and DI of N3 and N6 and the control treatments were analyzed following Duncan’s Multiple Range Test (DMRT). The F value detected the significance of treatments at *p* = 0.05.

The reduction of cassava leaf spot disease (RCLSD) is calculated based on DS with the formula (3):(3)$$\mathrm{RCLSD}\;(\%)=\frac{\mathrm{DS}\;\mathrm{of}\;\mathrm{negative}\;\mathrm{control}\;-\;\mathrm{DS}\;\mathrm{of}\;\mathrm{elicitors}}{\mathrm{DS}\;\mathrm{of}\;\mathrm{negative}\;\mathrm{control}}\;\times \;100$$

To assess their ability to maintain a stimulating effect on disease resistance, the cassava leaves without disease from the control groups were inoculated again with *A. alternata* H-Vi 7 conidia suspension (1 × 10^5^ conidia mL^−1^) at 63 DAP; N3 at 400 ppm and N6 at 200, 400 and 800 ppm were done in the same way. The DS and RCLSD were also calculated and analyzed as described above.

Furthermore, to evaluate their ability to enhance plant growth, the shoot height, the number of leaves at 28 and 42 DAP, the root length, the root weight, and the largest leaf area at 75 DAP of N3, N6, and the control treatments were recorded. Of these, the largest leaf area was measured by ImageJ 1.4 g software (National Institution of Health, Bethesda, MD, USA) (Figure 1).

## 3. Results

### 3.1. Synthesis and Characterization of CS-NP-Loaded SA and Ag

The hydrodynamic diameters of CS-NP-loaded SA (N3) and Ag (N6) were 89.86 ± 9.04 nm and 249.67 ± 23.97 nm, the PDIs were 0.36 ± 0.02 and 0.53 ± 0.03, and the zeta potentials were 22.27 ± 1.01 and 13.53 ± 0.74 mV, respectively (Figure 2). Under the FESEM, N3 and N6 have spherical forms and porous architecture (Figure 3). In the FTIR test, the peaks 3422 (NH_2_ stretch–primary amide), 1656 (CO-NH_2_–amide group), 1597 (NH_2_ bend–primary amide), and 897 (Anhydro glycoside) of bulk CS shifted to 3421, 1640, 1540, and 895 cm^−1^ in N3 and 3423, 1643, 1542, and 894 cm^−1^ in N6, respectively. This indicated the interactions of CS, TPP, and SA or Ag in N3 or N6, respectively. Moreover, the shift to 1314 cm^−1^ in N3 showed the interaction of COOH and NH_2_ (Figure 4). The interaction of function groups in N3 and N6 has shown success in the ionic gelation process.

### 3.2. Phytotoxicity Test

The phytotoxicity test was performed to evaluate the potential toxicity of NPs on cassava leaves. Usually, the necrotic spots or the browning around the leaves’ disks’ margins are the result of toxicity caused by chemicals in the leaf disk assay method. The larger the percentage of necrotic or browning area, the greater the toxicity. The results showed that the six NPs formulations with six concentrations did not cause necrotic spots or the browning around the leaves’ disks’ margins. The leaves’ disks of control groups did not show toxicity, except for the Pyr treatment, which turned the leaves’ disks to slight-yellow, and the SN treatment, which turned the leaves’ disks’ margins to brown—a positive toxicity (Figure 5). This confirmed that NP formulation does not cause toxicity on cassava leaves.

### 3.3. Screening Elicitor Formulations for Inducing Resistance of Cassava Leaf Spot Disease and Growth in Cassava Plants under Net-House Conditions

An overview of the experiment to evaluate the effect of CS-NP-loaded SA or Ag formulations on reducing leaf spot and enhance plant growth on cassava is shown in Figure 1.

Statistical analysis was performed on the DS and DI data of the six formulations (N1, N2, N3, N4, N5, and N6) by two factors, including formulation and concentration, to select an effective formulation among CS-NP-loaded SA and CS-NP-loaded Ag (Table 2 and Table 3). Overall, the DS and DI of CS-NP-loaded Ag were lower than CS-NP-loaded SA. The DS of N6 was significantly lower than the other formulations, at 8.54%. The DS of N3 was 10.83%; lower than N1 and N2, but not significantly different when compared with N4 (10.97%) and N5 (10.40%) (Table 2). The DI of N6 was 36.43%, which was also significantly lower than the other formulations, except for N4, at 40.49%. Of the CS-NP-loaded SA, the DI of N3 was 47.03%, which was significantly lower than N1 (57.71%) and N2 (58.82%) (Table 3). In addition, concentrations of 200 ppm and 400 ppm were more effective in reducing both DS and DI. Therefore, CS-NP-loaded SA (N3) and CS-NP-loaded Ag (N6) were selected for the subsequent analysis.

The DS and DI of N3 and N6 with six concentrations, including 25, 50, 100, 200, 400, and 800 ppm, were compared with the control group, including water, SA, CS, SN, Pyr, Flu, and ZON, according to the DMRT model to select the most effective concentration to calculate the reduction of leaf spot DS, based on this data. In general, the DS of the water treatment was significantly higher than the other treatments; in other words, they can reduce cassava leaf spot disease. The commercial fungicide NP (ZON) was used as a standard that reduces DS by 55.7%. The DS of CS, SN, and N3 at 50 ppm is significantly higher than ZON, which reduces DS by 39.4–52.1%. The DS of SA, Flu, and N3 at 100 ppm was non-significant when compared with ZON treatment. The DS of Pyr, N3 at 25, 200, and 800 ppm, and N6 at 25 ppm were significantly lower than ZON, which led to reducing the DS by 57.7–59.7%. Meanwhile, the DS of N3 at 400 ppm and N6 at 200, 400, and 800 ppm were significantly lower than ZON, reducing DS by 68.9–73.6%. The DI of the water treatment was 62.9%, which was significantly higher than the other treatments, except for the slightly significant CS treatment (52.1%) and N3 at 25 ppm (55.2%). The DI of N3 at 50 ppm and N6 at 25 ppm were non-significantly compared with ZON (44.4%). The DI of SA, CS, SN, N3 at 25, 100, and 200 ppm, and N6 at 50 ppm were significantly higher than ZON. Flu, N3 at 400 and 800 ppm, and N6 at 100 ppm and 400 ppm are similar, but with a decreasing trend. Meanwhile, the DI of N6 at 200 ppm and 800 ppm were 31.4 and 27.9%, respectively, which was significantly lower than ZON (Table 4, Figure 6). The N3 at 400 ppm and N6 at 200, 400, and 800 ppm were effective NP treatments that were selected to continue to inoculate *A. alternata* H-Vi 7 at 63 DAP (three weeks after the last spraying treatment) to assess maintenance effectiveness in stimulating disease resistance. In this test, the DS of water was also significantly higher than the other treatments, which indicated reductions in DS by 14.2–37.7%. The DS of Pyr was 33.9%, which non-significantly compared to ZON (34.1%). The DS of CS and SN was slightly significantly lower than ZON but tended to decrease. The DS of SA, Flu, N3 at 400 ppm, and N6 at 200, 400, and 800 ppm were significantly lower than ZON, which can reduce DS by 30.9–37.7%. Among them, the NPs treatments were slightly significantly lower than SA and Flu (Table 5, Figure 7).

Several concentrations of CS-NP-loaded SA or Ag were able to enhance cassava plant growth, including shoot height, the number of leaves, the number of shoots, the largest leaf area, root length, and root weight (Table 6). The N6 at 50 ppm treatment was superior, enhancing shoot height at 28 and 42 DAP, which increased by 40.7 and 28.2% when compared with the water treatment, respectively. However, the shoot height was inhibited by the N3 at 200 ppm and N3 at 50 ppm treatments at 28 and 42 DAP with 30.6 and 26.4%, respectively. Interestingly, N3 at 200 ppm increased shoot height by 69.4% when compared with the water treatment and was 35.8% within two weeks. In addition, N3 at 400 ppm and N6 at 200, 400, and 800 ppm appear to be non-significant in enhancing shoot height. Most of the other treatments significantly increased the number of leaves at 28 and 42 DAP when compared with the water treatment. The Pyr and ZON treatments were more effective in increasing the number of leaves by 79.2–88.7% and 103.9–123.5% at 28 and 42 DAP, respectively. The N3 and N6 treatments increased by 20.8–66.0% and 13.2–41.5 at 28 DAP, respectively. N3 at 25, 400, and 800 ppm was not significantly different when compared with Pyr and ZON. The N3 and N6 treatments increased by 43.1–82.4% and 45.1–86.3% at 42 DAP, respectively. N3 at 25 and 400 ppm and N6 at 50 and 200 ppm are not significantly different when compared with Pyr and ZON. Interestingly, the N6 at 50 and 200 ppm treatments increased the number of leaves by 46.2 and 46.8% within two weeks; that was the highest among the treatments, even Pyr and ZON. The treatments all increased the number of shoots when compared with the water treatment, except for the CS treatment. The N3 at 400 ppm and N6 at 50 ppm treatments were higher than the other treatments, increasing the number of shoots by 64.3 and 76.9% at 28 and 42 DAP, respectively. The N6 at 200, 400, and 800 ppm increased by 14.3% and 38.5–46.2% at 28 and 42 DAP, respectively. Interestingly, the Pyr, Flu, and ZON treatments increased by 28.6–50% and 46.2–61.5% at 28 and 42 DAP, respectively. The treatments, except for SN, significantly increased the largest leaf area when compared with the water treatment. The N3 and N6 treatments have increased the largest leaf area by 20.2–69.6% and 29.6–86.7%, respectively. In it, the N6 at 50 ppm was higher than the other treatments. N3 at 25, 200, and 800 ppm was non-significant when compared with ZON and CS. N3 at 400 ppm and N6 at 200, 400, and 800 ppm were increased by 28.8–41.9% compared with the water treatment, similar to SA, Pyr, and Flu. The root length and root weight were both increased by treatments when compared with the water treatment. However, a few treatments that did not increase significantly included the CS, SN, Pyr, and Flu treatments (root length) and the SA, CS, Pyr, Flu, N3, and N6 at 25 ppm treatments (root weight). The N3 and N6 treatments increased the root length by 11.6–27.4% and 20.1–31.7%, respectively, and the root weight by 10.3–27.6%, and 31.0–82.8%, respectively. Interestingly, the SN treatment increased root weight by 70.0%, which is similar to ZON (Figure 8).

## 4. Discussion

The hydrodynamic diameter of CS-NP-loaded SA (N3) was 89.86 ± 9.04 nm, which is smaller than the size of CS-NP-loaded SA in the study of [24], which was 368.7 nm. But the PDI and zeta potential of N3 are larger and smaller, respectively. Previously, CS-NP-loaded Ag was synthesized by the ionic gelation method with a size of 90.29 nm and a zeta potential of +92.05 mV with antibacterial properties that apply in medical (pharmaceutical) applications [29]. In plant disease management, CS-NP-loaded metals are usually Cu and Zn. In these studies, the DLS of CS-NP-loaded Cu was 295.4 nm, PDI 0.28, 19.6 mV [45]; 361.3 nm, PDI 0.2, 22.1 mV [46]; 314 nm, PDI 0.48, 19.5 mV [47]; 374.3 nm, PDI 0.33, 22.6 mV [48]; and 196.4 nm, PDI 0.5, +88 mV [25]. In addition, the DLS of CS-NP-loaded Zn was 387 nm, PDI 0.22, 34 mV [39]. NPs in the studies of [39,46,47,48] were used as elicitors. The CS-NP-loaded Ag (N6) with a size of 249.67 ± 23.97 nm was smaller than the size of NP in these studies. But the PDI and zeta potential of N3 are larger and smaller, respectively. This is also the first case study in plant disease management. The FTIR test showed the interaction of a primary amide, an amide group, and an anhydro glycoside group in CS-NP-loaded SA (N3) and CS-NP-loaded Ag (N6) formulations compared with CS bulk. The peaks 3422 (NH_2_ stretch), 1656 (CO-NH_2_), 1597 (NH_2_ bend), and 897 (Anhydro glycoside) of bulk CS shifted to 3421, 1640, 1540, and 895 cm^−1^ in N3 and 3423, 1643, 1542, and 894 cm^−1^ in N6, respectively. The shift to 1314 cm^−1^ in N3 showed an interaction of COOH and NH_2_. The FTIR peaks are different from previous studies, but still in the range that confirms successful synthesis [24,39,49].

Before an application on the cassava plant, the CS-NP-loaded SA and Ag were tested for phytotoxicity with cassava leaves by the leaf disk assay method. Previously, this method has also been used to determine the dose threshold for the toxicity of 8-methoxynaphthalen-1-ol—an antifungal compound on tomato—and (±)-botryodiplodin—a phytotoxin produced by *M. phaseolina* that causes charcoal rot disease on soybeans [40,41]. The results show that the CS-NPs formulations were not the cause of the necrotic spots or the browning around leaves’ disks’ margins. But the SN treatment turned the leaves’ disks’ margins brown (Figure 5). This confirmed that NP formulations have a potential non-toxicity to the cassava plant when compared with using a single chemical (SN). This allowed that NP formulations could be treated on the cassava plants. Recent studies also show that metal NPs (Ag, MgO, and ZnO) can directly affect a fungal pathogen, a bacterial pathogen, or both under in vitro, greenhouse, and field conditions [50,51,52]. Interestingly, the effectiveness of reducing the incidence of soft rot disease and enhancing sugar beet growth and sucrose content by Ag NP treatment was higher than that of *Bacillus subtilis* or algal extract [50]. Furthermore, MgO NP was able to inhibit spore germination, sporangium formation, and hyphal development of *Phytophthora nicotianae* and *Thielaviopsis basicola*, as well as reduce tobacco black shank and black root rot disease with an efficacy control reaching 50.20% and 62.10%, respectively [51]. These CS-NPs continue to further research on the following experiments in net-house conditions.

The net-house experiment was conducted to select one CS-NP-loaded SA and one CS-NP-loaded Ag as effective formulations with effective concentrations to reduce cassava leaf spot disease as an elicitor (pre-treating before pathogen infection). Through three rounds of statistical analysis of DS and/or DI data, N3 at 400 ppm and N6 at 200, 400, and 800 ppm were shown as high-potential elicitors (Figure 1). Specifically, six formulations with six concentrations were statistically analyzed by two factors, which showed that CS-NP-loaded SA (N3) and CS-NP-loaded Ag (N6) were effective formulations. The results of the second statistical analysis showed that N3 at 400 ppm and N6 at 200, 400, and 800 ppm were effective concentrations to reduce cassava leaf spot disease by 68.9, 72.9, 73.6, and 73.2%, respectively. Therefore, they were selected for further pathogen inoculation at 63 DAP (three weeks after the last spraying treatment). The results showed that they were also effective in reducing disease by 37.0, 37.4, 37.0, and 37.7%, respectively. Commercial fungicides also have the ability to reduce cassava leaf spot disease. Of these, pyraclostrobin (Pyr) was more effective than flutriafol (Flu) at 56 DAP (59.6 and 54.7%, respectively) but lower at 75 DAP (14.7 and 35.4%, respectively). They control cassava brown leaf spot disease by spraying fungicide at 10 DAI. A study by [5] also showed that flutriafol was more effective than pyraclostrobin in reducing the area under the disease progress curve by 68.7 and 11.3%, respectively. In addition, a commercial NP (ZON) only reduces 55.7 and 14.2% at 56 and 75 DAP, respectively. That is significantly lower than the effective concentrations of N3 and N6. In enhancing cassava growth, the effective concentration usually did not significantly increase the shoot height (−13.4–9.1%), but the opposite was true for the number of leaves (45.1–82.4%) and shoots (38.5–46.2%). These treatments significantly increased the largest leaf area (29.6–41.9%), root length (11.6–29.9%), and root weight (27.6–82.8%). The commercial NP usually has a higher potential to enhance plant growth because the active ingredient is ZnO, which contains an important nutrient element (Zn) for plant growth. Overall, N3 and N6 treatments took the spotlight in reducing leaf spot disease and enhancing plant growth on cassava in this study.

In this study, the CS-NP-loaded SA or Ag formulations were used as elicitors that can induce plants’ defense systems against pathogens. In previous studies, CS-NPs (0.05–0.1%) have been reported to reduce finger millet blast [53], wheat Fusarium head blight [54], and rice sheath blight [55] by inducing POD, PAL, chitinase, ROS, superoxide activity, and H_2_O_2_ content in plants. In addition, CS-NPs have been loaded with active ingredients, including Harpin protein [56], Cu [46], Zn [39], SA [24], and thiamine [57], that can induce plants’ defense systems against infections of *R. solani* (tomato), *C. lunata* (maize), *F. verticillioides* (maize), and *F. oxysporum* (chickpea) by inducing CAT, chitinase, PAL, PO, POD, PPO, protease, SOD, 𝛽-1,3-glucanase activity, O_2_^-^, H_2_O_2_ content, and lignin localization. Moreover, the CS-NP-loaded Cu also reduced bacterial pustule disease in soybeans by 40.6–49.7%, but interestingly, a low concentration (0.06%) was more effective than a high concentration (0.16%) [47]. In our study, N3 at 400 ppm was more effective than N3 at 800 ppm, and N3 at 25 ppm was equally effective as N3 at 100, 200, and 800 ppm. A mixture of CS-NPs (ionic gelation method) and Cu-NPs (chemical reduction method) can also reduce vascular wilt disease in date palms by 16.2–59.3% by inducing total phenol, phenoloxidase, and POD [58].

The formulations contain CS, SA, or Ag, so their effectiveness in disease reduction may be due to the synergistic effect of CS and SA or Ag (Figure 9). This is the direct effect of the NP formulations. NPs can also be good carriers to transport CS, SA, or Ag into plant cells, leading to an indirect effect of increasing plants’ resistance against plant diseases.

The plant’s innate immune system has three stages: perception, signal transduction, and defense response. This process could be induced by an elicitor [10].

The CS can act as an elicitor that activates a plant’s innate immunity, including stimulating H_2_O_2_ production, nitric oxide, generating PR protein, oxidative burst, enzymes, callose, and secondary metabolite (phytoalexin, suberin, lignin, and phenolic compound) [21,59,60]. In the [61] study, CS treatment (2.5 mg/mL) increased plant height (39%), stem girth (44%), and reduced powdery mildew (*Erysiphe cichoracearum*) disease (66.6%) on cucumbers. Furthermore, this effect was associated with an increase in the production of benzyl aminopurine, indole acetic acid, 1-napthol acetic acid and lignin, callose and H_2_O_2_, PPO, PAL, POD, and glucanase in the plant.

SA is a plant hormone that plays a role in plant germination, growth, and immunity. In the cell, endogenous SA is produced by the PAL and isochorismate pathways. When the SA concentration is increased, the cellular reduction potential is changed, and the NPR1 structure changes to a monomer that can enter the nucleus. Here, NPR1 binds to specific TGA transcription factors and then expresses defenses against pathogen attacks [62,63]. SA treatment can increase a plant’s defense system in chickpeas, including enzymes (POD and PPO), total phenol, H_2_O_2_, and protein content [64]. Previously, [65] reported that exogenous SA treatment (1 mM) was able to reduce bacterial leaf blight in rice by 38% with an increase in superoxide anion production and hypersensitive response, as well as lignin and pectin content in the cell wall. In addition, the formatted SA (Zacha11 at 500 mg/L) has increased stem height, root length, and the number of roots, and reduced root rot disease by 53.33% on cassava [66].

Silver nitrate also has effects on plants, depending on its concentration. MS medium containing silver nitrate (1 mg/L) could increase rooted shoots (63.6%), plant height (78.6%), number of roots (181.3%), and root length (508%), and reduce bacterial contamination in *Gentiana lutea* tissue culture [67]. Furthermore, MS medium containing silver nitrate (1 mg/L) could improve the quality of shoots and decrease the time required for rooting to earlier than two months in two cultivars of *Anthurium andraeanum* under tissue culture [68]. In addition, [69] compared the effectiveness of silver nitrate and Ag-NP (green synthesis) on the growth of rice under biotic stress conditions. The results showed that Ag-NP was more effective than silver nitrate at the same 75 mg/L concentration in increasing root length (1.2 and 12.8%), shoot length (21 and 20%), root number (8.1 and 6.8%), fresh weight (6.4 and 5%), dry weight (4.6 and 3.5%), leaf area (58.5 and 57.2%), leaf number (4.3 and 3.7%), leaf fresh weight (1.7 and 1.4%), and leaf dry weight (0.9 and 0.8%) under *Aspergillus* infection. Furthermore, the aflatoxins of Ag-NP were 3.5 ± 0.1 µg/kg compared to silver nitrate, which was 3.9 ± 0.3 µg/kg. In [70]’s study, Ag-NP (60 ppm) also increased common bean and maize growth, including shoot length (47.0 and 27.9%), root length (56.1 and 46.1%), fresh weight (85.9 and 109.2%), dry weight (74.4 and 122.0%), leaf area (56.5 and 70.0%), chlorophyll a (49.0 and 46.0%), chlorophyll b (33.0 and 26.0%), and carbohydrate content (57.0 and 62.0%). However, at higher concentrations (100 ppm), growth was reduced. In addition, Ag-NP (50 ppm) treatment in lilies resulted in increased plant height (7.6%), number of leaves (27.2%), greenness index (17.6%), leaf fresh weight (35.1%), bulb fresh weight (73.4%), and number of scales (24.3%), but at higher concentrations the effect is not equivalent [71].

**Figure 9 polymers-14-00660-f009:**
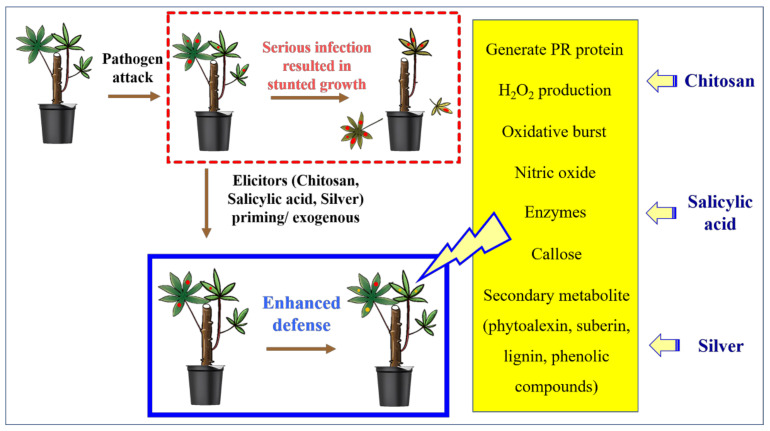
The effect of CS, SA, and Ag on plants’ defenses. Modified from [21,59,60,61,62,63,64,65,66,69].

Why are NP formulations highly effective in reducing plant diseases and enhancing plant growth? In general, the preeminent characteristics of NPs are their small size, large contact surface area, and high reactivity, leading to their applications in controlling disease and enhancing plant growth [72]. NPs can be absorbed by plants through foliar, brand, trunk, and root [73]. CS-NPs with a nano size and a positive charge are able to easily penetrate cells or stick to plant surfaces [21]. CS-NP can enter the plant via leaves (the stomata and cuticular pathway) and roots (the diffusion and cuticular pathway). The stomata of cassava leaves are from 18.2–24.9 µm × 12.1–16.1 µm [74]. The hydrodynamic diameters of N3 and N6 were 89.86 ± 9.04 and 249.67 ± 23.97 nm, while the PDIs were 0.36 ± 0.02 and 0.53 ± 0.03 and the zeta potentials were 22.27 ± 1.01 and 13.53 ± 0.74 mV, respectively. Therefore, these NPs can pass through the stomata or be easily absorbed by the cassava plant. CS-NP can adjust osmotic pressure in the cell, resulting in increased uptake and availability of water and nutrients [75]. In addition, when sticking to plants, the CS-NPs loaded with active ingredients including Hexaconazole, Zn, Cu, SA, Harpin protein, NPK, and silicon can slowly release their active ingredients so that plants can absorb them slowly, as reported in the studies [24,39,46,56,76,77,78]. CS is commonly used as a carrier due to its solubility in aqueous media and its ability to mix with organic, inorganic, or copolymer compounds to increase solubility [79]. The main component of CS is nitrogen, so the carrier (CS) can act as a source of nitrogen for plants to absorb, or enhance cell division, cell elongation, enzymatic activation, and synthesis of protein, which leads to increased yields [21,80].

## 5. Conclusions

In this study, three formulations of CS-NP-loaded SA (N1, N2, and N3) and three formulations of CS-NP-loaded Ag (N4, N5, and N6) were synthesized by the ionic gelation method. The leaf disk assay method confirmed that CS-NP formulations up to 800 ppm do not cause toxicity on cassava leaves. The CS-NPs formulations at 25–800 ppm were mainly applied as elicitors. The results showed that the CS-NP-loaded SA (N3) and CS-NP-loaded Ag (N6) were more effective than the remaining other formulations (N1, N2, N4, and N5) in reducing the disease severity and the disease index of leaf spot. The N3 at 400 ppm and N6 at 200, 400, and 800 ppm could reduce disease severity by 68.9–73.6% and 37.0–37.7%, respectively, when a fungal pathogen was inoculated at 2 and 21 days after spraying elicitors with a density of 10^4^ and 10^5^ conidia per mL. This was more effective than commercial zinc oxide NP (ZON), which was 55.7 and 14.2%, respectively, pyraclostropin (Pyr) fungicide, which was 59.6 and 14.7%, respectively, and flutriafol (Flu) fungicide, which was 54.7 and 35.4%, respectively. The N3 and N6 treatments also enhanced cassava growth, including shoot height, the number of leaves, the number of shoots, the largest leaf area, the root length, and the root weight, with a similar effect to the positive control group. That indicates the potential to use CS-NP-loaded SA or Ag at low concentrations (200–400 ppm) as elicitors to manage cassava leaf spot disease. The limitation of the ionic gelation method is that it is difficult to create NPs that have uniform sizes, resulting in unstable efficiency. The effect of the size of the NP on plant disease management also varied between reports. To solve this problem, this study was approached with a focus on reverse research models from preparing elicitors, toxicity tests, screening formulation effectiveness to concentration effectiveness, and the characterization of effective elicitors (Figure 10). This research motif focuses first on the effectiveness in in vivo (net-house) conditions after preparing elicitors and toxicity tests instead of in vitro conditions. Finally, only the highly effective formulations of CS-NP-loaded SA (N3) and CS-NP-loaded Ag (N6) were characterized, instead of focusing on all six formulations from the start. In addition, the NP formulations in this research were focused for use as elicitors at low concentrations. Therefore, the effective concentrations of NP formulation do not necessarily inhibit fungal pathogens, which are often concentrated in in vitro experiments. This reduces the time and cost of the research process. A field experiment (200–400 ppm) is still needed before it is widely recommended. This research motif and CS-NPs-based ionic gelation method also hold promise in plant or agricultural disease management, especially in areas where modern techniques are limited.

## Figures and Tables

**Figure 1 polymers-14-00660-f001:**
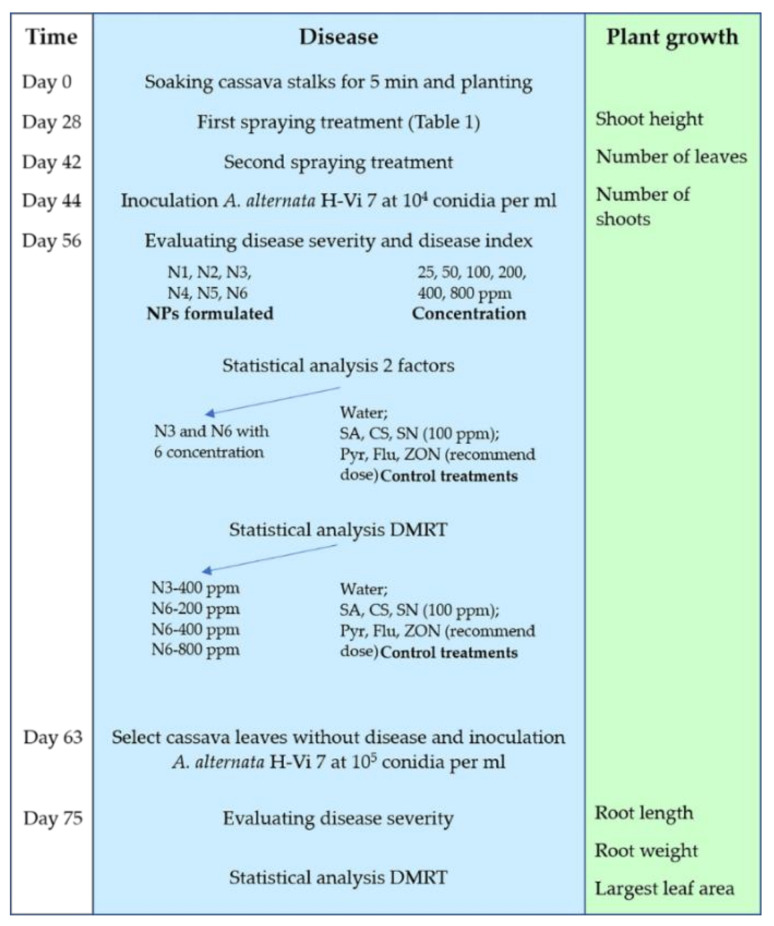
An overview of the “Screening elicitor formulations for inducing resistance to cassava leaf spot disease and growth in cassava plants under net-house conditions” experiments.

**Figure 2 polymers-14-00660-f002:**
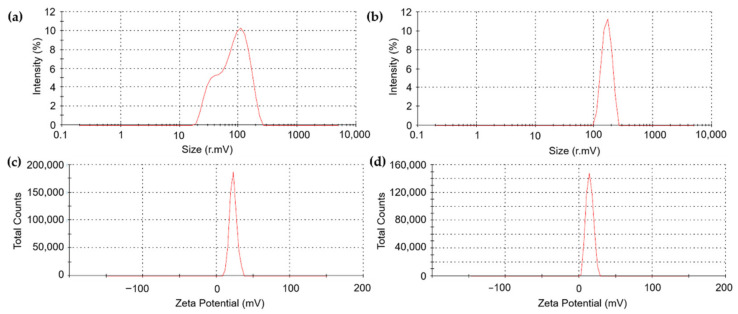
The DLS analyses of CS-NP-loaded SA’s (N3) (**a**) size and (**c**) zeta potential and CS-NP-loaded Ag’s (N6) (**b**) size and (**d**) zeta potential.

**Figure 3 polymers-14-00660-f003:**
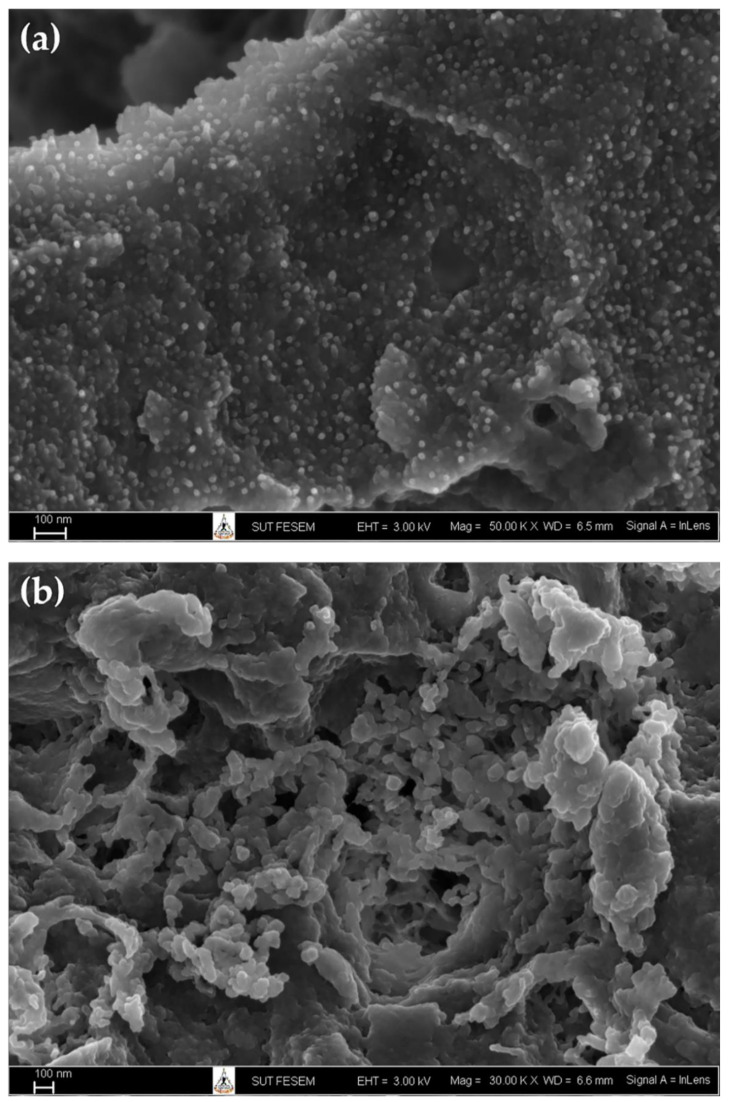
The morphology of (**a**) CS-NP-loaded SA (N3) and (**b**) CS-NP-loaded Ag (N6) formulations under a Field Emission Scanning Electron Microscope.

**Figure 4 polymers-14-00660-f004:**
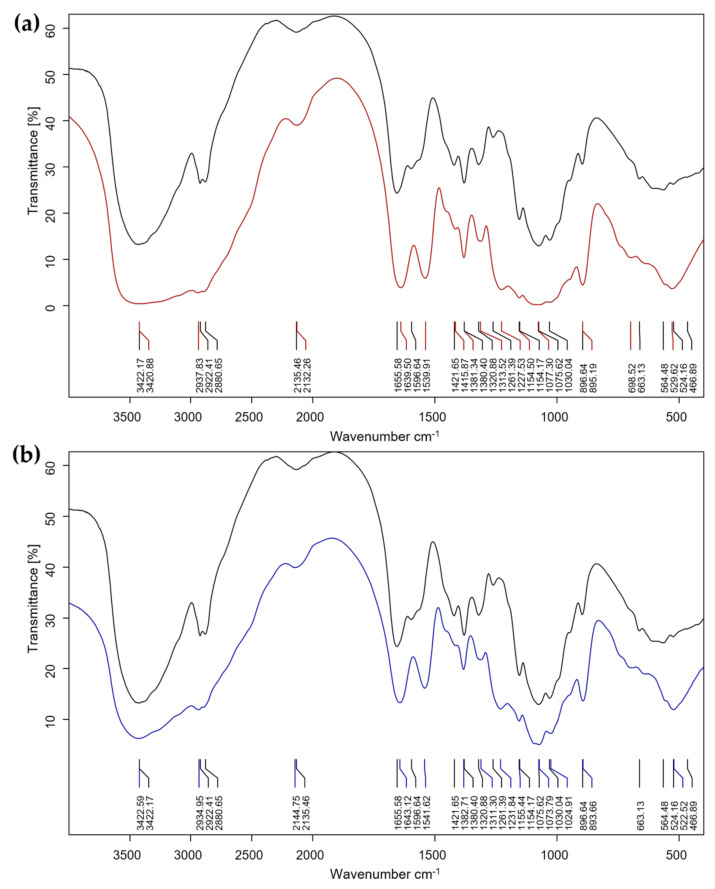
The FTIR analysis of CS (black) compared with (**a**) CS-NP-loaded SA (N3-red) and (**b**) CS-NP-loaded Ag (N6-blue).

**Figure 5 polymers-14-00660-f005:**
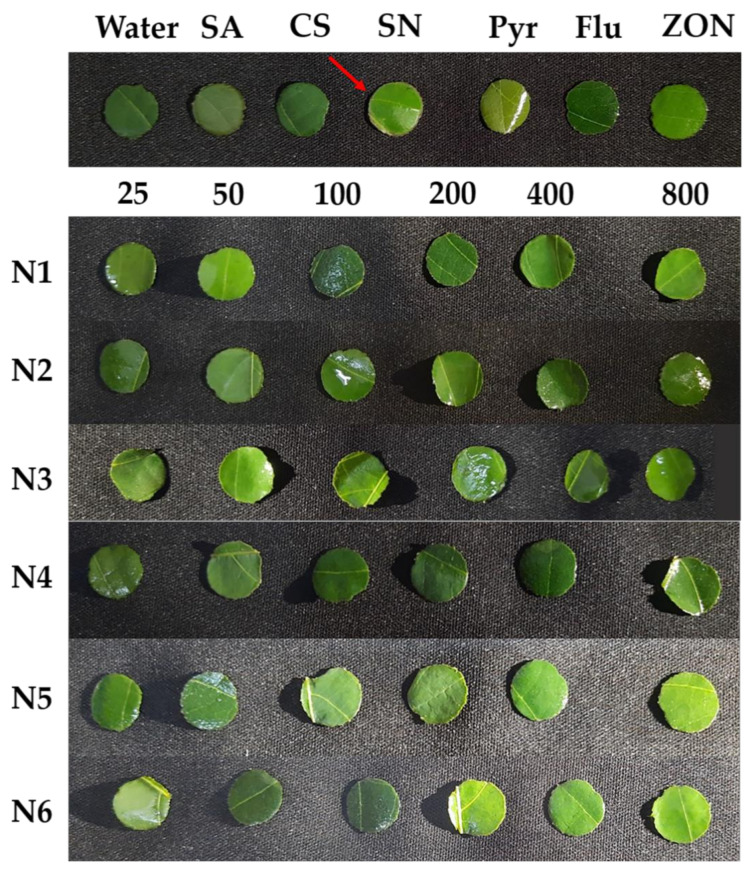
The phytotoxicity tests of the six NP formulations and the control groups by leaf disk assay. Note: SA-Salicylic acid 100 ppm; CS-Chitosan 100 ppm; SN-Silver nitrate 100 ppm; Pyr-Pyraclostrobin, Headline^®^ 10 mL/20 L; Flu-Flutriafol, JOINT^®^ 30 mL/20 L; ZON-Zinc oxide NP, ZONO-S1^®^ 20 mL/20 L. 1 The CS (0.4 or 0.5%) and Pentasodium triphosphate (0.2 or 0.5%) were mixed with SA varying at 0.05, 0.1 and 0.2% or silver nitrate varying at 1, 2, and 3 mM to prepare three formulations of CS-NP-loaded SA, named N1, N2, and N3, or three formulations of CS-NP-loaded Ag named N4, N5, and N6, respectively.

**Figure 6 polymers-14-00660-f006:**
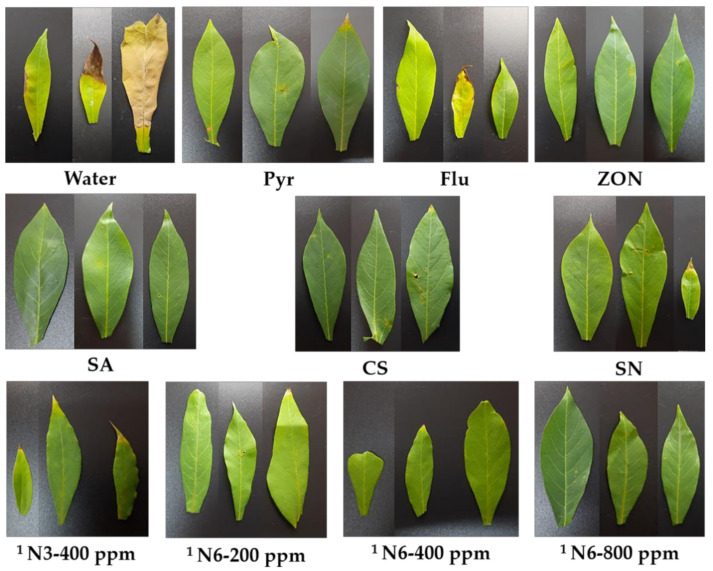
The symptoms of cassava infected by *A. alternaria* H-Vi 7 at 56 DAP. Note: SA-Salicylic acid 100 ppm; CS-Chitosan 100 ppm; SN-Silver nitrate 100 ppm; Pyr-Pyraclostrobin, Headline^®^ 10 mL/20 L; Flu-Flutriafol, JOINT^®^ 30 mL/20 L; ZON-Zinc oxide NP, ZONO-S1^®^ 20 mL/20 L. ^1^ The CS (0.4% or 0.5%) and Pentasodium triphosphate (0.2% or 0.5%) were mixed with SA 0.2% or silver nitrate 3 mM to prepare a formulation of CS-NP-loaded SA (N3) or a formulation of CS-NP-loaded Ag (N6), respectively.

**Figure 7 polymers-14-00660-f007:**
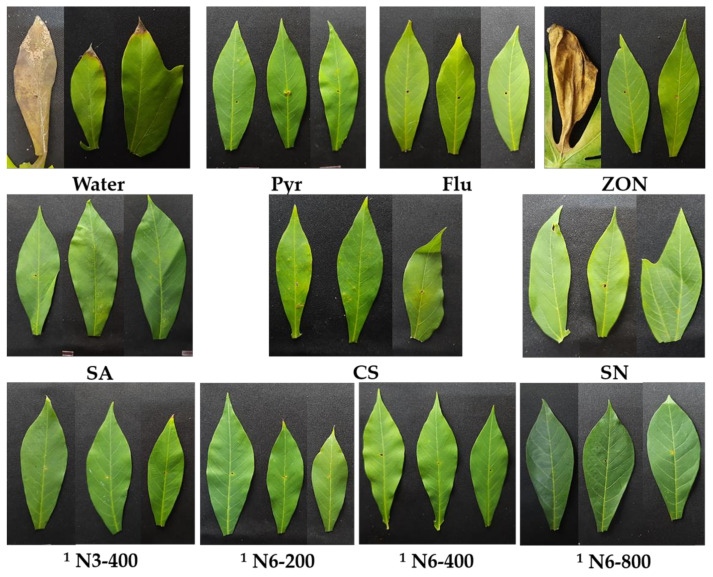
The symptoms of cassava infected by *A. alternaria* H-Vi 7 at 75 DAP. Note: SA-Salicylic acid 100 ppm; CS-Chitosan 100 ppm; SN-Silver nitrate 100 ppm; Pyr-Pyraclostrobin, Headline^®^ 10 mL/20 L; Flu-Flutriafol, JOINT^®^ 30 mL/20 L; ZON-Zinc oxide NP, ZONO-S1^®^ 20 mL/20 L. ^1^ The CS (0.4 or 0.5%) and Pentasodium triphosphate (0.2 or 0.5%) were mixed with SA 0.2% or silver nitrate 3 mM to prepare a formulation of CS-NP-loaded SA (N3) or a formulation of CS-NP-loaded Ag (N6), respectively.

**Figure 8 polymers-14-00660-f008:**
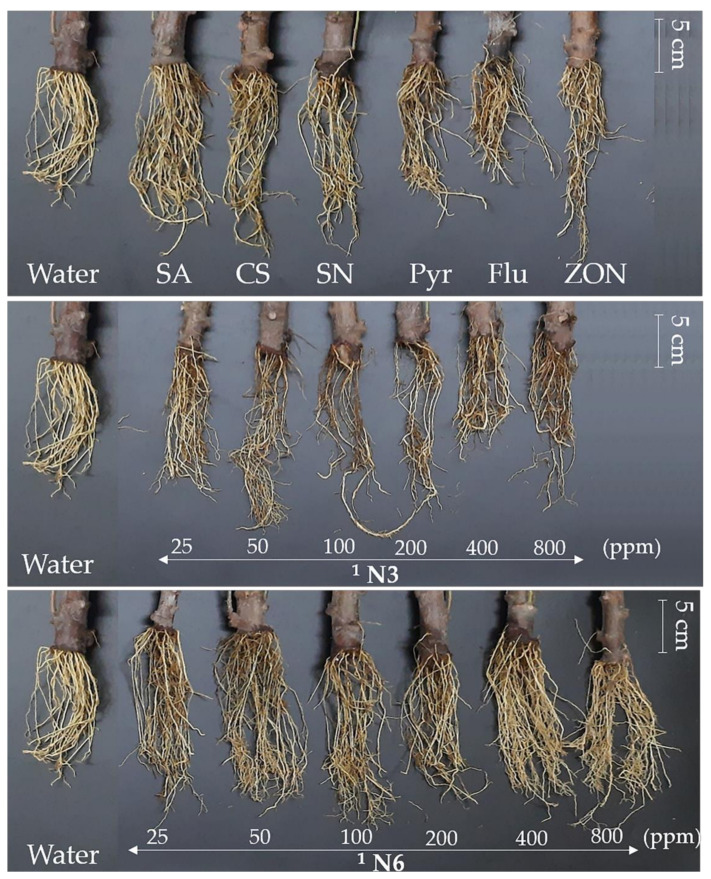
The roots of the cassava plants at 75 DAP. Note: SA-Salicylic acid 100 ppm; CS-Chitosan 100 ppm; SN-Silver nitrate 100 ppm; Pyr-Pyraclostrobin, Headline^®^ 10 mL/20 L; Flu-Flutriafol, JOINT^®^ 30 mL/20 L; ZON-Zinc oxide NP, ZONO-S1^®^ 20 mL/20 L. 1 The CS (0.4 or 0.5%) and Pentasodium triphosphate (0.2 or 0.5%) were mixed with SA 0.2% or silver nitrate 3 mM to prepare a formulation of CS-NP-loaded SA (N3) or a formulation of CS-NP-loaded Ag (N6), respectively.

**Figure 10 polymers-14-00660-f010:**
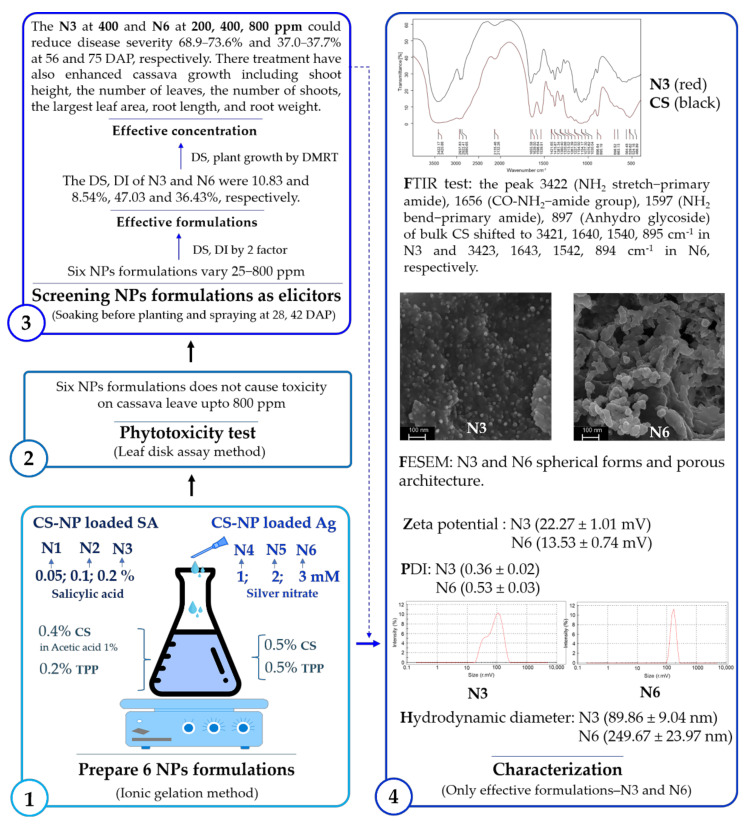
The graphical scheme of the complete study and highlighted results that were approached with a focus on a reverse research model from (1) preparing elicitors, (2) toxicity tests, (3) screening formulation effectiveness to concentration effectiveness, and (4) the characterization of effective elicitors.

**Table 1 polymers-14-00660-t001:** The treatments used in the study.

Treatments	Concentrations	Note
3 formulations of CS-NP-loaded SA (named N1, N2, N3)	25, 50, 100, 200, 400, 800 ppm	NP elicitor group
3 formulations of CS-NP-loaded Ag (named N4, N5, N6)		
SA (Salicylic acid), CS (Chitosan) and SN (Silver nitrate)	100 ppm	Single chemical group
Pyr (Pyraclostrobin, Headline^®^)	10 mL/20 L (recommend dose)	Positive control group
Flu (Flutriafol, JOINT^®^)	30 mL/20 L (recommend dose)	
ZON (Zinc oxide NP, ZONO-S1^®^)	20 mL/20 L (recommend dose)	
Water	100 ppm	Negative control group

**Table 2 polymers-14-00660-t002:** The disease severity (%) of cassava leaf spot at 56 days after planting, statistically analyzed by two factors.

Nano Particles ^1^	Concentration (ppm)						Mean ^2^
	25	50	100	200	400	800	
N1	9.66	17.19	14.90	13.19	9.47	16.99	13.57 c
N2	11.84	16.11	18.13	17.54	18.10	16.16	16.31 d
N3	10.43	15.20	11.04	10.38	7.81	10.11	10.83 b
N4	11.88	10.86	11.31	8.96	10.93	11.88	10.97 b
N5	10.56	17.01	10.57	10.04	7.39	6.81	10.40 b
N6	10.61	11.86	8.61	6.80	6.62	6.72	8.54 a
Mean ^3^	10.83 AB	14.70 D	12.43 C	11.15 AB	10.05 A	11.44 BC	11.77
CV (%)	16.67						

^1^ The CS (0.4% or 0.5%) and Pentasodium triphosphate (0.2% or 0.5%) were mixed with SA varying at 0.05%, 0.1%, and 0.2% or silver nitrate varying at 1, 2, and 3 mM to prepare three formulations of CS-NP-loaded SA named N1, N2, and N3 or three formulations of CS-NP-loaded Ag named N4, N5, and N6, respectively. ^2,3^ Means followed by the same letter do not differ significantly according to DMRT at *p* ≤ 0.01.

**Table 3 polymers-14-00660-t003:** The disease index (%) of cassava leaf spot at 56 days after planting, statistically analyzed by two factors.

Nano Particles ^1^	Concentration (ppm)						Mean ^2^
	25	50	100	200	400	800	
N1	36.5	75.0	70.0	53.1	44.1	67.5	57.71 d
N2	44.9	58.1	65.3	61.8	61.6	61.2	58.82 d
N3	55.2	45.2	48.3	48.5	43.0	42.0	47.03 c
N4	37.6	41.3	41.5	39.1	38.4	45.0	40.49 ab
N5	38.8	59.2	46.8	40.2	33.8	28.7	41.26 b
N6	45.0	46.2	34.9	31.4	33.2	27.9	36.43 a
Mean ^3^	43.00 A	54.15 B	51.16 B	45.68 A	42.35 A	45.38 A	46.95
CV (%)	16.49						

^1^ The CS (0.4% or 0.5%) and Pentasodium triphosphate (0.2% or 0.5%) were mixed with SA varying at 0.05%, 0.1%, and 0.2% or silver nitrate varying at 1, 2, and 3 mM to prepare three formulations of CS-NP-loaded SA named N1, N2, and N3 or three formulations of CS-NP-loaded Ag named N4, N5, and N6, respectively. ^2,3^ Means followed by the same letter do not differ significantly according to DMRT at *p* ≤ 0.01.

**Table 4 polymers-14-00660-t004:** The disease severity, reduction of disease severity, and disease index of cassava leaf spot of N3 and N6 treatments at 56 days after planting compared with control treatments.

Treatments ^1^	Disease Severity ^2^ (%)	Reduction of Disease Severity (%)	Disease Index ^2^ (%)
Control	25.1 f	-	62.9 h
SA 100 ppm	10.9 cd	56.7	51.1 fg
CS 100 ppm	12.0 d	52.1	52.1 f–h
SN 100 ppm	12.3 d	50.8	47.6 e–g
Pyr	10.1 b–d	59.6	38.0 a–e
Flu	11.4 cd	54.7	41.7 b–f
ZON	11.1 cd	55.7	44.4 c–g
N3-25 ppm	10.4 b–d	58.4	55.2 gh
N3-50 ppm	15.2 e	39.4	45.2 c–g
N3-100 ppm	11.0 cd	56.0	48.3 e–g
N3-200 ppm	10.4 b–d	58.6	48.5 e–g
N3-400 ppm	7.8 ab	68.9	43.0 b–g
N3-800 ppm	10.1 b–d	59.7	42.0 b–f
N6-25 ppm	10.6 b–d	57.7	45.0 c–g
N6-50 ppm	11.9 d	52.8	46.2 d–g
N6-100 ppm	8.6 a–c	65.7	34.9 a–d
N6-200 ppm	6.8 a	72.9	31.4 ab
N6-400 ppm	6.6 a	73.6	33.2 a–c
N6-800 ppm	6.7 a	73.2	27.9 a
F-test	**		**
CV (%)	16.41		16.87

^1^ SA-Salicylic acid 100 ppm; CS-Chitosan 100 ppm; SN-Silver nitrate 100 ppm; Pyr-Pyraclostrobin, Headline^®^ 10 mL/20 L; Flu-Flutriafol, JOINT^®^ 30 mL/20 L; ZON-Zinc oxide NP, ZONO-S1^®^ 20 mL/20 L. The CS (0.4% or 0.5%) and Pentasodium triphosphate (0.2% or 0.5%) were mixed with SA 0.2% or silver nitrate 3 mM to prepare a formulation of CS-NP-loaded SA (N3) or a formulation of CS-NP-loaded Ag (N6), respectively. ^2^ Means followed by the same letter do not differ significantly according to DMRT at *p* ≤ 0.01 (**).

**Table 5 polymers-14-00660-t005:** The disease severity and reduction of disease severity of cassava leaf spot at 75 days after planting.

Treatments ^1^	Disease Severity ^2^ (%)	Reduction of Disease Severity (%)
Control	39.7 e	
SA 100 ppm	27.4 a–c	30.9
CS 100 ppm	32.0 cd	19.3
SN 100 ppm	30.3 b–d	23.8
Pyr	33.9 d	14.7
Flu	25.6 ab	35.4
ZON	34.1 d	14.2
N6-200	24.8 a	37.4
N6-400	25.0 a	37.0
N6-800	24.7 a	37.7
N3-400	25.0 a	37.0
F-test	**	
CV (%)	10.87	

^1^ SA-Salicylic acid 100 ppm; CS-Chitosan 100 ppm; SN-Silver nitrate 100 ppm; Pyr-Pyraclostrobin, Headline^®^ 10 mL/20 L; Flu-Flutriafol, JOINT^®^ 30 mL/20 L; ZON-Zinc oxide NP, ZONO-S1^®^ 20 mL/20 L. The CS (0.4% or 0.5%) and Pentasodium triphosphate (0.2% or 0.5%) were mixed with SA 0.2% or silver nitrate 3 mM to prepare a formulation of CS-NP-loaded SA (N3) or a formulation of CS-NP-loaded Ag (N6), respectively. ^2^ Means followed by the same letter do not differ significantly according to DMRT at *p* ≤ 0.01 (**).

**Table 6 polymers-14-00660-t006:** The plant growth parameters enhanced by N3 and N6 formulation compared with the control treatments.

Treatments ^1^	Shoot Height ^2^ (cm)		The Number of Leaves ^2^		The Number of Shoots ^2^		Largest Leaf Area ^2^ (cm^2^)	Root Length ^2^ (cm)	Root Weight ^2^ (g)
	28 DAP	42 DAP	28 DAP	42 DAP	28 DAP	42 DAP			
Control	8.1 bc	11.0 b–e	5.3 f	5.1 d	1.4 de	1.3 d	48.0 e	16.4 g	2.9 f
SA 100 ppm	7.0 cd	10.9 b–e	7.4 b–e	8.2 bc	1.6 a–e	1.6 b–d	62.2 cd	20.1 a–e	3.8 b–f
CS 100 ppm	8.0 bc	11.5 b–d	5.6 ef	7.3 cd	1.2 e	1.2 d	77.8 b	18.1 e–g	4.0 b–f
SN 100 ppm	7.3 cd	11.7 b–d	6.5 d–f	7.2 cd	1.6 b–e	1.6 b–d	48.7 e	16.8 fg	4.9 a–c
Pyr	8.8 b	12.7 ab	9.5 a	10.4 ab	2.1 a–c	2.1 ab	65.1 c	17.1 fg	3.7 c–f
Flu	7.4 b–d	12.7 ab	9.1 ab	8.6 bc	2.1 ab	1.9 a–c	67.9 c	17.4 fg	3.3 ef
ZON	8.9 b	12.3 a–c	10.0 a	11.4 a	1.8 a–e	1.9 a–c	82.8 b	19.0 b–f	4.6 a–d
N3-25 ppm	8.0 bc	11.3 b–e	8.7 a–c	9.3 a–c	1.9 a–d	1.8 a–d	77.2 b	20.0 a–e	3.7 d–f
N3-50 ppm	8.2 bc	8.7 f	6.4 ef	7.6 c	1.6 b–e	1.6 b–d	61.5 cd	18.9 c–f	3.2 ef
N3-100 ppm	8.9 b	11.9 b–d	6.6 d–f	8.1 bc	1.4 de	1.6 b–d	57.7 d	20.9 a–c	3.5 d–f
N3-200 ppm	6.2 d	10.5 c–f	6.9 c–f	7.3 cd	1.7 a–e	1.6 b–d	76.6 b	19.8 a–e	3.5 d–f
N3-400 ppm	7.9 bc	12.0 a–c	8.8 a–c	9.3 a–c	2.3 a	1.9 a–c	62.3 cd	18.3 d–g	3.7 d–f
N3-800 ppm	7.6 bc	12.6 a–c	8.4 a–d	8.8 bc	1.6 a–e	1.8 a–d	81.4 b	20.1 a–e	3.5 d–f
N6-25 ppm	8.8 b	12.0 a–c	6.0 ef	8.3 bc	1.4 c–e	1.5 b–d	67.7 c	20.6 a–d	3.8 b–f
N6-50 ppm	11.4 a	14.1 a	6.5 d–f	9.5 a–c	1.5 b–e	2.3 a	89.6 a	19.8 a–e	4.2 a–e
N6-100 ppm	7.1 cd	9.3 ef	7.4 b–e	8.5 bc	1.5 b–e	1.6 b–d	64.0 cd	21.6 a	4.2 a–e
N6-200 ppm	7.9 bc	11.7 b–d	6.2 ef	9.1 a–c	1.6 b–e	1.9 a–c	64.3 cd	19.7 a–e	4.3 a–e
N6-400 ppm	8.3 bc	9.7 d–f	6.6 d–f	7.4 cd	1.6 a–e	1.9 a–c	68.1 c	21.3 ab	5.0 ab
N6-800 ppm	8.2 bc	10.8 b–e	7.5 b–e	7.6 c	1.6 a–e	1.8 a–d	62.2 cd	20.4 a–d	5.3 a
F-test	**	**	**	**	*	*	**	**	**
CV (%)	10.71	11.14	16.43	17.67	23.24	22.20	6.63	7.21	18.36

^1^ SA-Salicylic acid 100 ppm; CS-Chitosan 100 ppm; SN-Silver nitrate 100 ppm; Pyr-Pyraclostrobin, Headline^®^ 10 mL/20 L; Flu-Flutriafol, JOINT^®^ 30 mL/20 L; ZON-Zinc oxide NP, ZONO-S1^®^ 20 mL/20 L. The CS (0.4% or 0.5%) and Pentasodium triphosphate (0.2% or 0.5%) were mixed with SA 0.2% or silver nitrate 3 mM to prepare a formulation of CS-NP-loaded SA (N3) or a formulation of CS-NP-loaded Ag (N6), respectively. ^2^ Means followed by the same letter do not differ significantly according to DMRT at *p* ≤ 0.01 (**) or 0.01 < *p* ≤ 0.05 (*); DAP: days after planting.

## Data Availability

Not applicable.

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
