# Peer review of "Efficacy of Chitosan Nanoparticle Loaded-Salicylic Acid and -Silver on Management of Cassava Leaf Spot Disease"

_polymers, 2022, doi:10.3390/polym14040660_

Round 1

Reviewer 1 Report

The manuscript "Efficacy of chitosan nanoparticle loaded-salicylic acid and -sil-2 version management cassava leaf spot disease" describes the results of a socially significant study on a method of combating a disease that reduces the yield of cassava, a valuable agricultural plant. The authors used a new approach to the problem, which consists in finding a safer method than the use of existing fungicides. The article is written satisfactorily, the methods given are reproducible. The way the material is presented could be improved for ease of understanding.

The study can be accepted after minor revision.

Line 45. Phytoplasma are bacteria, so they should not have been mentioned separately.

Line 69. It is recommended to add links to modern sources, such as https://doi.org/10.3390/mi12121480 .

Lines 146-147. The sentence is out of context.

Lines 204-206. The sentence is out of context.

Line 402. Apparently, I should have written "fungal" instead of "fugal".

Figure 5 shows a phytotoxicity test. It is not entirely clear why drilled leaf discs were used instead of whole leaves. The separation of a part of the leaf is a rather serious intervention, which can have additional consequences by changing the effects of chemicals.

Chapter 4 Discussion is overloaded with information, including numerical data. It is desirable to discuss the results of the study in a more understandable way.

Chapter 5 Conclusions, by contrast, contains little resulting data. It would be interesting to place a little more positions in it, summarizing everything written earlier.

Reviewer 2 Report

In Introduction safety aspects of the use of nanotechnologies should be also mentioned and related references added such as:

Zielińska et al. Nanotoxicology and Nanosafety: Safety-By-Design and Testing at a Glance. Int J Environ Res Public Health. 2020 Jun 28;17(13):E4657. doi: 10.3390/ijerph17134657. PMID: 32605255.

The aim of paper should be better expressed and the novelty character should be better marked.

A graphical scheme of complete study approach should be inserted.

Lines 203-206 should be checked. The authors should add some introductory lines to better introduce and describe the different types of results.

Figures on FTIR should be greatly implemented and also the description of FTIR results in the text.

"3.2. Phytotoxicity test" should be better described.

Check legend of Table 4.

Limits, advantages, practical applications and future direction should be added in Conclusion.

The authors should check that the paper is formatted following journal guidelines. Check typos along the document.

Size and resolutions of Figures and Tables should be uniforme along the paper.
